# Iron Transporter Protein Expressions in Children with Celiac Disease

**DOI:** 10.3390/nu13030776

**Published:** 2021-02-27

**Authors:** Marleena Repo, Markus Hannula, Juha Taavela, Jari Hyttinen, Jorma Isola, Pauliina Hiltunen, Alina Popp, Katri Kaukinen, Kalle Kurppa, Katri Lindfors

**Affiliations:** 1Tampere Centre for Child Health Research, Tampere University and Tampere University Hospital, 33014 Tampere, Finland; marleena.repo@tuni.fi (M.R.); pauliina.hiltunen@pshp.fi (P.H.); kalle.kurppa@tuni.fi (K.K.); 2Celiac Disease Research Center, Faculty of Medicine and Health Technology, Tampere University, 33520 Tampere, Finland; katri.kaukinen@tuni.fi; 3Faculty of Medicine and Health Technology and BioMediTech Institute, Tampere University, 33520 Tampere, Finland; markus.hannula@tuni.fi (M.H.); jari.hyttinen@tuni.fi (J.H.); 4Central Finland Central Hospital, 40620 Jyväskylä, Finland; juha.taavela@tuni.fi; 5Laboratory of Cancer Biology, Faculty of Medicine and Health Technology, Tampere University, 33520 Tampere, Finland; jorma.isola@tuni.fi; 6Jilab Inc, 33520 Tampere, Finland; 7National Institute for Mother and Child Health, Carol Davila University of Medicine and Pharmacy, 050474 Bucharest, Romania; alina.popp@tuni.fi; 8Department of Internal Medicine, Tampere University Hospital, 33521 Tampere, Finland; 9Department of Pediatrics, Seinäjoki Central Hospital and University Consortium of Seinäjoki, 60320 Seinäjoki, Finland

**Keywords:** celiac disease, anemia, iron transporter

## Abstract

Anemia is a frequent finding in children with celiac disease but the detailed pathophysiological mechanisms in the intestine remain obscure. One possible explanation could be an abnormal expression of duodenal iron transport proteins. However, the results have so far been inconsistent. We investigated this issue by comparing immunohistochemical stainings of duodenal cytochrome B (DCYTB), divalent metal transporter 1 (DMT1), ferroportin, hephaestin and transferrin receptor 1 (TfR1) in duodenal biopsies between 27 children with celiac disease and duodenal atrophy, 10 celiac autoantibody-positive children with potential celiac disease and six autoantibody-negative control children. Twenty out of these 43 subjects had anemia. The expressions of the iron proteins were investigated with regard to saturation and the percentage of the stained area or stained membrane length of the enterocytes. The results showed the stained area of ferroportin to be increased and the saturation of hephaestin to be decreased in celiac disease patients compared with controls. There were no differences in the transporter protein expressions between anemic and non-anemic patients. The present results suggest an iron status-independent alteration of ferroportin and hephaestin proteins in children with histologically confirmed celiac disease.

## 1. Introduction

Celiac disease is an immune-mediated disorder driven by ingested gluten [1]. A frequent and sometimes the only clinical finding in untreated patients is anemia, generally considered to be caused by damaged duodenal mucosa and the resulting malabsorption of iron [2,3]. Nevertheless, there is a poor correlation between the presence of anemia and the severity of histological damage [2,4,5]. Moreover, duodenal absorption of only about 10% of the dietary iron fulfills the daily needs [6], indicating that the reduced mucosal surface area is not the sole explanation for anemia. In fact, it may be present in so-called potential celiac disease, referring to subjects with endomysial (EmA) or transglutaminase 2 (TGA) celiac autoantibodies but with a normal small bowel morphology [5,7,8,9], suggesting that the pathophysiologic mechanisms behind iron deficiency and anemia are more complex than previously thought. 

In healthy conditions, iron is absorbed from the gut by a sophisticated and tightly regulated process [6,10]. In the apical membrane of enterocytes, the duodenal cytochrome B (DCYTB) reduces iron to a ferrous form. A divalent metal transporter (DMT1) transfers ferrous iron into the enterocyte where it is either utilized in mitochondria, stored as ferritin or transported to the circulation via basolateral ferroportin. Before being able to bind to the plasma iron carrier transferrin, iron must be reconverted into a ferric form by basolateral hephaestin. The enterocytes may also reuptake iron for their own metabolic functions through transferrin receptor 1 (TfR1). A key regulator of iron absorption and metabolism is hepcidin, which reduces the iron uptake in enterocytes and its release from body storages [11,12]. The details of this regulation, however, are not fully understood [13,14,15,16]. 

It has been suggested that the abnormal expression of the iron transporter proteins could provide an explanation for anemia in celiac disease. So far only a few studies have tested this hypothesis with inconsistent findings [17,18,19,20]. We therefore aimed to investigate possible altered transporter protein expression by staining the DMT, DCYTB, ferroportin, hephaestin and TfR1 in duodenal biopsies of children with histologically confirmed or potential celiac disease and autoantibody-negative controls.

## 2. Materials and Methods

### 2.1. Patients and the Study Design

The study was conducted at Tampere University Hospital, Tampere, Finland and the National Institute for Mother and Child Health, Bucharest, Romania. Twenty-seven children (age < 17 years) with EmA and/or TGA and a duodenal lesion comprised the celiac disease group. Ten children with positive EmA and TGA but a non-diagnostic histology comprised the potential celiac disease group. Six children who were endoscopied due to unexplained gastrointestinal symptoms but who had normal duodenal villi and negative EmA/TGA were used as controls. All 43 children were further divided into those with or without anemia.

The study was conducted according to the Helsinki Declaration. The study protocol and patient recruitment were approved by the Ethics Committee of the Pirkanmaa Hospital District, Finland and the Ethics Committees of the University of Medicine and Pharmacy “Carol Davila” and the National Institute for Mother and Child Health “Alessandrescu-Rusescu”, Romania. Written informed consent was obtained from all study participants and their guardians.

### 2.2. Celiac Disease Serology and Small Bowel Mucosal Morphology

EmA titers were measured by an indirect immunofluorescence method using a human umbilical cord as a substrate [21]. A dilution of 1:5 was considered positive and positive sera were further diluted 1:50, 1:100, 1:200, 1:500, 1:1000, 1:2000 and 1:4000. The EliA Celikey test (Phadia, Uppsala, Sweden) was used to determine TGA. The cut-off for seropositivity was set at >7.0 U/L according to the manufacturer’s instructions.

A minimum of four representative forceps biopsies were taken from the duodenum. The paraffin-embedded specimens were cut, stained with hematoxylin and eosin and evaluated for celiac disease diagnosis by an experienced pathologist. Only correctly oriented histological sections were accepted for the histological analyses [22]. Subjects with crypt hyperplasia and a villous atrophy in the duodenal mucosa (Corazza–Villanacci B1-B2) were diagnosed with celiac disease whereas children with a non-diagnostic histology (Corazza–Villanacci A) formed the potential celiac disease and control groups [23,24]. 

### 2.3. Laboratory Parameters and Hepcidin

The following associated laboratory parameters were measured by standard methods: hemoglobin (reference value (Rf) from 100–141 to 130–160 g/L depending on age and sex [25]), plasma soluble transferrin receptor (sTfR, Rf from 1.6–5.2 mg/L to 2.0–6.8 mg/L), mean corpuscular volume (MCV; Rf from 72–88 to 87–146 fl [25]), serum total iron (Fe, Rf 6–25 mmol/L), plasma ferritin (Rf > 10 mg/L), transferrin iron saturation (Rf 15–50%), serum folate (Rf 10.4–42.4 nmol/L) and serum vitamin B12 (Rf 140–490 pmol/L). In addition, serum bioactive hepcidin (hepcidin-25) levels were measured using a commercial solid-phase enzyme-linked immunosorbent assay (EIA-5258, DRG Diagnostics, Marburg, Germany) according to the manufacturer’s instructions [5].

### 2.4. Immunohistochemistry

For the immunohistochemistry, 5 µm-thick sections were cut from the formalin-fixed, paraffin-embedded duodenal specimens. After deparaffination and rehydration antigens were exposed by heat-induced epitope retrieval. Thereafter, a non-specific staining was blocked followed by overnight incubation with primary antibodies (Appendix A). After washing the primary antibodies, the specimens were incubated overnight with a secondary antibody prior to the blocking of the endogenic peroxidase and a visualization of the staining with either ImmPRESS or VECTASTAIN Elite ABC reagent (Vector Laboratories Inc, Peterborough, UK). Finally, sections were counterstained with hematoxylin. 

### 2.5. Digital Analysis of the Stained Sections

All slides were scanned as whole-slide images using a SlideStrider scanner at a resolution of 0.16 µm per pixel (Jilab Inc., Tampere, Finland). The images were stored as JPX files and viewed with a JVSview program from where they were exported to a Fiji Image J program for further analysis [26]. Of the DCYTB sections, both the entire visible epithelial apical membrane and the DCYTB stained membrane were drawn and measured. The stained membrane length was divided by the whole membrane length to assess the percentage of the apical membrane covered with the protein. Thereafter, from DMT1, ferroportin, hephaestin and TfR1 stained sections of the epithelium were selected, other parts cut out and the images consisting of only the epithelium were stored as TIF files (Appendix A). Subsequently, the files were transferred to a Matlab program (The MathWorks Inc. Natick, Massachusetts) where they were transformed from RGB to HSV images to access the color saturation independently of the lightness. To measure only the primary antibody staining, a red color was chosen from the hue channel within values 0–0.1 and 0.9–1. The saturation channel was then thresholded according to all sections in each stained protein series using Otsu’s method [27]. Finally, the value of the mean saturation of each section divided by the maximum saturation of the protein series and percentage of the stained area were measured for each section. 

### 2.6. Statistical Analysis

All statistical analyses were performed using IBM SPSS Statistics version 26.0 (IBM Corp. Armonk, NY). The clinical characteristics and prevalence of anemia are presented as percentage distributions. The skewness of the quantitative data was assessed by the Shapiro–Wilk method and most of the variables were not normally distributed. For simplicity, all data are thus expressed as medians with quartiles except for age, which is given with a median and a range. Staining results as mean/maximum saturation and the stained area were compared between groups using a non-parametric Mann–Whitney U test. Correlations between hepcidin, plasma transferrin receptor 1, serum ferritin and the DCYTB stained apical border percent and in other proteins’ mean/maximum saturations and stained areas were calculated using Spearman’s rank (rS) correlation. *p* values < 0.05 were considered significant.

## 3. Results

There was no significant difference between children with celiac disease and potential celiac disease in age, gender or median hepcidin values or, despite a non-significant trend, in the frequency of anemia or low MCV (Table 1). The former group nevertheless had a higher frequency of increased sTfR values and lower ferritin (Table 1) as well as a higher median EmA (1:1000 vs 1:50, *p* < 0.001) and TGA (120 U/l vs. 17 U/l, *p* = 0.001). The controls (two boys, two girls, 50% anemia) were slightly older (median 10.6 (range 3.3, 15.3) years) than the celiac and potential celiac patients.

The stained area of ferroportin was increased in the celiac disease patients compared with the controls and a similar although non-significant trend was observed in the saturation of the staining (Table 2). In hephaestin the saturation was significantly decreased in celiac disease compared with the controls with a similar trend in the stained area. No significant differences between the study groups were observed in either saturation or the stained area of the other iron transporters (Table 2), nor were there any differences in either the saturation or the stained area of any of the iron transporters between children with or without anemia (Table 3).

There was a positive correlation between ferritin values and TfR1 saturations (r_S_ 0.594, *p* = 0.015) and the stained area (r_S_ 0.761, *p* = 0.001) in children with celiac disease. A moderate negative correlation was also found between sTfR values and hephaestin saturation (r_S_ –0.349, *p* = 0.046) when evaluated in all study subjects whereas this was not observed when evaluated separately in celiac disease patients. No other correlations between the hepcidin, ferritin or sTfR values and the stainings of the iron transporter were detected (data not shown).

## 4. Discussion

The main finding of the present study was an increased expression of ferroportin and a decreased expression of hephaestin in children with histologically confirmed celiac disease compared with the non-celiac controls. There were no other significant differences between the study groups in the expression of iron transporter proteins. In addition, no differences in any of these proteins were detected when anemic and non-anemic children were evaluated separately.

The expression of the iron transporter proteins and/or their coding mRNAs in celiac disease have previously been reported in three studies comprising adult patients and in one pediatric study [17,18,19,20]. In line with our results, Sharma et al. showed an iron status-independent increase in protein levels of ferroportin but also of DMT1 in untreated adult celiac disease [17]. Additionally, they found increased DMT1 and ferroportin mRNAs in iron deficient celiac disease patients and also in anemic non-celiac controls. Tolone et al. later reported that DMT1 mRNA was increased in celiac disease children with mild but not with severe atrophy compared with controls with normal duodenal mucosa [20]. However, they included both potential celiac disease patients and suspected gastroesophageal reflux disease patients in the control group. Additionally, Matysiak-Budnik reported an upregulation of TfR1 protein levels in adults with untreated celiac disease [19]. Barisani et al. reported increased mRNAs and protein levels of DMT1, ferroportin, hephaestin and TfR1 in adult celiac disease patients but, in contrast to the protein levels in ours and Sharma’s studies, these findings were iron status-dependent [18]. However, unlike others, Barisani et al. included both untreated patients and patients on a gluten-free diet in the celiac disease group. No earlier studies have reported the decreased hephaestin expression observed here. 

These partially inconsistent results between the studies may be attributable to the differences in the number and clinical characteristics of the participants and/or by the variable use of primary antibodies and staining protocols. On the other hand, there may in fact be significant differences between children and adults in intestinal iron transporter protein expression [28]. As our results lacked major outliers and were also consistent within and between the study groups, we believe the present findings to reflect the true state of iron transporter protein expression in the duodenal mucosa of children with untreated celiac disease. 

Our findings would suggest that changes in ferroportin and hephaestin expression do not explain the intestinal pathophysiology of anemia in celiac disease but may rather reflect the immaturity of the epithelium [29] of the atrophic duodenal mucosa. Interestingly, Tolone et al. found a distinct polymorphism in the DMT1 gene to be significantly more frequent in anemic than in non-anemic children with celiac disease; in fact, the polymorphism conferred a four-fold risk for the development of anemia [20]. Furthermore, a polymorphism in the transmembrane serine protease 6 gene can be overrepresented in celiac disease patients and its presence predicts an inadequate response to iron supplementation [30,31] whereas polymorphisms in the human hemochromatosis protein gene may provide protection against anemia in celiac disease [31,32,33]. Thus, genetic variants affecting iron metabolism may at least partially determine a predisposition to anemia in celiac disease.

As an additional novel finding of the present study, we observed a moderately positive correlation between the TfR1 saturation and stained area and the serum ferritin levels in children with celiac disease. Additionally, a negative correlation between the saturation of hephaestin and sTfR levels was shown among all of the children although this was not seen in celiac disease patients when evaluated separately. As sTfR usually increases and ferritin decreases in subjects with iron deficiency, an opposite correlation pointing towards a compensatory increase of intestinal iron absorption would have been expected [34]. However, both the origin and function of circulating ferritin and sTfR are currently unknown [10] and thus their connection with the duodenal iron transporters needs to be further studied. 

## 5. Conclusions

To conclude, the iron status-independent changes observed here in ferroportin and hephaestin in children with histologically confirmed celiac disease likely reflect the immature nature of the epithelium in the atrophic disease state and do not explain the intestinal pathophysiology of anemia in children with celiac disease. Further investigations with a larger number of study subjects and in both children and adults are needed to understand the complex mechanisms of abnormal iron metabolism leading to anemia in celiac disease.

## Figures and Tables

**Table 1 nutrients-13-00776-t001:** Clinical characteristics and laboratory values of 37 children with celiac disease (CD) and potential CD.

Variable	CD, *n* = 27		Potential CD, *n*=10		*p* Value
	*n*	%	*n*	%	
Girls	18	67	8	80	0.431
Anemia	14	52	3	30	0.236
High sTfR	12	46	1	10	0.043
Low MCV	10	35	1	10	0.140
	Median	Q_1_, Q_3_	Median	Q_1_, Q_3_	
Age, yrs (range)	6.8	2.7, 14.4	6.1	4.1, 16.9	0.555
Ferritin, mg/L	7.0	4.8, 15.5	20.5	11.3, 29.8	0.017
Hepcidin, ng/mL	13.7	12.6, 15.2	15.4	13.2, 18.2	0.286

MCV, mean corpuscular volume; Q1 and Q3, lower and upper quartiles; sTfR, soluble transferrin receptor. Data was available from all cases except 126 and 217. ^1^26 and ^2^27

**Table 2 nutrients-13-00776-t002:** Iron transporter protein saturations and the stained areas of enterocytes in the duodenal biopsies of the study subjects.

Iron Transporter Protein	CD *N* = 27		Potential CD *N* = 10		Controls *N* = 6		CD vs. Potential CD	CD vs. Controls	Potential CD vs. Controls
	Median	Q_1_, Q_3_	Median	Q_1_, Q_3_	Median	Q_1_, Q_3_	*p* Value	*p* Value	*p* Value
DCYTB									
Stained apical border, %	54	36, 76	50	24, 79	50	33, 73	0.679	0.751	0.662
DMT1									
Mean/max saturation, %	42	36, 51	43	35, 52	37	33, 50	0.999	0.342	0.828
Stained area, %	59	56, 62	60	49, 67	57	48, 65	0.827	0.653	0.745
Ferroportin									
Mean/max saturation, %	64	62, 66	64	59, 69	61	59, 63	0.827	0.072	0.329
Stained area, %	66	54, 75	68	40, 78	45	22, 57	0.999	0.024	0.129
Hephaestin									
Mean/max saturation, %	27	25, 29	28	26, 31	31	27, 37	0.234	0.028	0.195
Stained area, %	1	0, 22	4	1, 21	16	8, 38	0.266	0.080	0.195
TfR1									
Mean/max saturation, %	52	48, 54	50	49, 55	53	51, 62	0.821	0.325	0.233
Stained area, %	59	49, 69	42	33, 68	64	47, 73	0.257	0.437	0.233

CD, celiac disease; DCYTB, duodenal cytochrome B; DMT1, divalent metal transporter 1; TfR1, transferrin receptor 1. Q_1_, Q_3_, upper and lower quartiles. Data available in each analysis were from at least 90% of the patients.

**Table 3 nutrients-13-00776-t003:** Iron transporter protein saturations and the stained areas of enterocytes in the duodenal biopsies of children with and without anemia.

Iron Transporter Protein	All Study Children, *n* = 43					Children With CD, *n* = 27				
	Anemia, *n* = 20		No Anemia, *n* = 23		*p* Value	Anemia, *n* = 14		No Anemia, *n* = 13		*p* Value
	Median	Q_1_, Q_3_	Median	Q_1_, Q_3_		Median	Q_1_, Q_3_	Median	Q_1_, Q_3_	
DCYTB										
Stained apical border, %	54	13, 78	56	37, 73	0.999	53	10, 79	63	42, 70	0.689
DMT1										
Mean/max saturation, %	43	37, 51	39	36, 54	0.582	43	39, 51	39	37, 53	0.446
Stained area, %	59	56, 62	59	54, 66	0.388	59	56, 61	59	57, 63	0.744
Ferroportin										
Mean/max saturation, %	64	59, 65	64	60, 68	0.372	64	62, 65	65	62, 69	0.128
Stained area, %	65	46, 74	65	44, 77	0.875	65	55, 74	66	51, 77	0.624
Hephaestin										
Mean/max saturation, %	27 ^1^	26, 29	28	25, 32	0.594	27 ^2^	25, 29	27	25, 31	0.663
Stained area, %	5 ^1^	1, 22	3	0, 23	0.795	3 ^2^	0, 19	1	0, 25	0.744
TfR1										
Mean/max saturation, %	50 ^1^	49, 54	52	49, 55	0.452	50 ^3^	49, 54	53	48, 55	0.750
Stained area, %	55 ^1^	42, 62	61	43, 70	0.292	55 ^3^	55, 64	61	50, 70	0.469

CD, celiac disease; DCYTB, duodenal cytochrome B; DMT1, divalent metal transporter 1; TfR1, transferrin receptor 1. Q_1_, Q_3_, upper and lower quartiles. Data available in each analysis were from at least 90% of the patients except ^1^ 17, ^2^ 12 and ^3^ 11 patients.

## Data Availability

Due to the protection of patient privacy, the original data used to support the findings of this study cannot be shared.

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
