# Peer review of "Iron Transporter Protein Expressions in Children with Celiac Disease"

_nutrients, 2021, doi:10.3390/nu13030776_

Round 1
Reviewer 1 Report
The authors have investigated iron transport protein expressions in children with celiac disease. This is an interesting and solid study. I have no methodological concerns.
Author Response
Thank you for the positive comments!
Reviewer 2 Report
We appreciate the Authors’work; we think that the article is well organized, clear and quite understandable.
In the article, authors compared the expression of iron proteins in children with celiac disease and controls. The aim of the paper is interesting because it suggest that the atrophy of the intestinal mucosa and the anemia can be connected to an abnormal expression of these proteins. They analyzed 43 children, 27children of which had celiac disease, 10 children with antibodies but negative histology were the potential celiac disease grup and 6 children were the controls. All of them had been bioptized and the histological findings analyzed. The results don’t show any difference between the case and the controls, so the study aim isn’t achieved.
Author Response
Thank you for the positive comments! It is true that there was no differences between the groups, which was somewhat unexpected and indicates that further studies on this issues are warranted. We have now emphasized this in page 8, lines 232-235.
Reviewer 3 Report
The work is clear and straightforward. The topic is dealt with in an exhaustive manner and the aim that was to be achieved is clearly expressed.
For the sector, the topic is interesting, although it does not lead to clear conclusions, perhaps due to the small number of subjects. It is certainly very complex, in the pediatric field, to reach a greater number of subjects with biopsy.
Correct the bibliographic notes:
number 5. For the name Lähdeaho, remove the “?” and enter a “ä”. Remove as last name Mäki M.
number 16. Year of publication is 2008 and not 2007. Add after volume 294 G192-8.
Author Response
Thank you for the positive comments! It is true that there was no differences between the groups, which was somewhat unexpected and could be explained by the small number of subjects. Therefore, further studies with a larger number of subjects on this issue are needed. We have now emphasized this in page 8, lines 232-235.
The unfortunate misspellings in the references have now been corrected. See page 8, lines 272-274 and page 9, lines 298 and 299.